# Filovirus receptor NPC1 contributes to species-specific patterns of ebolavirus susceptibility in bats

Melinda Ng[1†], Esther Ndungo[1†], Maria E Kaczmarek[2†], Andrew S Herbert[3], Tabea Binger[4], Ana I Kuehne[3], Rohit K Jangra[1], John A Hawkins[5], Robert J Gifford[6], Rohan Biswas[1], Ann Demogines[7], Rebekah M James[3], Meng Yu[8], Thijn R Brummelkamp[9], Christian Drosten[4,10], Lin-Fa Wang[8], Jens H Kuhn[11], Marcel A Müller[4], John M Dye[3*], Sara L Sawyer[7,12,13*], Kartik Chandran[1*]

[1]Department of Microbiology and Immunology, Albert Einstein College of Medicine, Bronx, United States; [2]Department of Integrative Biology, University of Texas at Austin, Austin, United States; [3]United States Army Medical Research Institute of Infectious Diseases, Fort Detrick, Frederick, United States; [4]Institute of Virology, University of Bonn Medical Center, Bonn, Germany; [5]Institute for Computational Engineering and Sciences, University of Texas at Austin, Austin, United States; [6]University of Glasgow MRC Virology Unit, Glasgow, United Kingdom; [7]Department of Molecular Biosciences, University of Texas at Austin, Austin, United States; [8]Program in Emerging Infectious Diseases, Duke-NUS Graduate Medical School, Singapore; [9]Netherlands Cancer Institute, Plesmanlaan, The Netherlands; [10]German Centre for Infectious Diseases Research, Bonn, Germany; [11]Integrated Research Facility at Fort Detrick, National Institute for Allergy and Infectious Diseases, National Institutes of Health, Fort Detrick, Frederick, United States; [12]BioFrontiers Institute, University of Colorado Boulder, Boulder, United States; [13]Department of Molecular, Cellular and Developmental Biology, University of Colorado Boulder, Boulder, United States

*For correspondence: john.m.
dye1.civ@mail.mil (JMD);
ssawyer@colorado.edu (SLS);
kartik.chandran@einstein.yu.edu
(KC)

†These authors contributed
equally to this work

Competing interests: The
authors declare that no
competing interests exist.

Reviewing editor: Richard A
Neher, Max Planck Institute for
Developmental Biology,
Germany

**Abstract** Biological factors that influence the host range and spillover of Ebola virus (EBOV) and other filoviruses remain enigmatic. While filoviruses infect diverse mammalian cell lines, we report that cells from African straw-colored fruit bats (*Eidolon helvum*) are refractory to EBOV infection. This could be explained by a single amino acid change in the filovirus receptor, NPC1, which greatly reduces the affinity of EBOV-NPC1 interaction. We found signatures of positive selection in bat *NPC1* concentrated at the virus-receptor interface, with the strongest signal at the same residue that controls EBOV infection in *Eidolon helvum* cells. Our work identifies *NPC1* as a genetic determinant of filovirus susceptibility in bats, and suggests that some *NPC1* variations reflect host adaptations to reduce filovirus replication and virulence. A single viral mutation afforded escape from receptor control, revealing a pathway for compensatory viral evolution and a potential avenue for expansion of filovirus host range in nature.

## Introduction

Ebola virus (EBOV) and some of its relatives in the family *Filoviridae* (filoviruses) cause sporadic outbreaks of a highly lethal disease. These outbreaks are thought to be initiated by viral spillover from

**eLife digest** Ebola virus and other filoviruses can cause devastating diseases in humans and other apes. Numerous small outbreaks of Ebola virus disease have occurred in Africa over the past 40 years. However, in 2013–2015, the largest outbreak on record took place in three Western African nations with no previous history of the disease.

Human outbreaks of Ebola virus disease likely begin when a person encounters an infected wild animal. Though it remains unclear precisely which animals harbor Ebola virus between outbreaks, and how they transmit the virus to humans or other primates, recent work showed that some filoviruses do infect specific types of bats in nature.

Ng, Ndungo, Kaczmarek et al. sought to identify the genes that influence whether or not a type of bat is susceptible to infection by Ebola virus and other filoviruses. Several filoviruses, including Ebola virus, were tested to see if they could infect cells that had been collected from four types of African fruit bats. These bats are all found in areas where outbreaks have occurred in the past.

The tests revealed that a small change in the sequence of the *NPC1* gene in some bat cells greatly reduced their susceptibility to Ebola virus. *NPC1* encodes a protein that mammals need in order to move cholesterol within their cells. In humans, the loss of the protein encoded by *NPC1* causes a rare but very severe disease called Niemann-Pick type C disease. This protein also turns out to be a receptor that the filoviruses must bind to before they can infect the cells. Further analysis then revealed that *NPC1* has evolved rapidly in bats, with changes concentrated in the parts of the receptor that interact with Ebola virus.

Ng, Ndungo, Kaczmarek et al. went on to discover some changes in the genome sequence of Ebola virus that could compensate for the changes in the bat's *NPC1* gene. These findings hint at one way that a filovirus could evolve to better infect a host with receptors that were less than optimal.

Following on from this work, the next challenges will be to expand the investigation to include additional types of bats, other types of mammals, and other host genes that could influence filovirus infection and disease. Further studies could also examine the other side of the arms race – that is, the evolution of viral genes in bats. However, such studies would be complicated by the lack of viral sequences that have been collected from bats, because to date most have been isolated from humans and other primates instead.

an animal reservoir to a highly susceptible accidental host, such as a human or nonhuman primate (*Feldmann and Geisbert, 2011*; *Leroy et al., 2005*; *Towner et al., 2009*). Recent work suggests that some filoviruses infect bats in nature, and that these viruses may be distributed more widely than previously recognized. Very short RNA fragments corresponding to portions of ebolavirus genomes were detected in several frugivorous bats of the family Pteropodidae ('Old World fruit bats') in both Africa and Asia (*Leroy et al., 2005*; *Jayme et al., 2015*), and longer filovirus RNA fragments and near-complete RNA genomes were isolated from insectivorous Schreibers's long-fingered bats in Asia and Europe, respectively (*Negredo et al., 2011*; *He et al., 2015*). However, despite considerable efforts, infectious ebolaviruses have never been recovered from bats. By contrast, Marburg (MARV) and Ravn (RAVV) viruses were found to circulate in Egyptian rousettes (*Rousettus aegyptiacus*), indicating that these bats are susceptible to MARV/RAVV and encounter them frequently in nature. Egyptian rousettes have been proposed as natural hosts for these viruses (*Amman et al., 2012*; *Towner et al., 2009*). This progress notwithstanding, many key questions remain. For example, the biological factors that influence filovirus host range and interspecies transmission are still poorly understood, as are the virus-host relationships that determine which species of bats are susceptible to infection by EBOV and other filoviruses.

Viral entry receptors are key determinants of tissue tropism and host range (*Radoshitzky et al., 2008*; *Sheahan et al., 2008*; *Hueffer et al., 2003*; *Demogines et al., 2013*). Niemann-Pick C1 (NPC1), a highly conserved endo/lysosomal protein involved in cellular cholesterol trafficking, was recently identified to be an essential entry receptor for all known filoviruses (*Côté et al., 2011*; *Carette et al., 2011*; *Miller et al., 2012*; *Ng et al., 2014*). In this study, we uncover a pattern of virus and host species specificity in the filovirus susceptibility of bat cells, which can be explained by

changes in the affinity of the essential interaction between NPC1 and the filovirus entry glycoprotein, GP. Crucially, genetic analyses reveal that *NPC1* is under positive selection in bats, with a strong signature of selection at precisely the same residue that influences the filovirus-receptor interaction. Our findings suggest that amino acid sequence changes in NPC1 at these positively-selected sites represent host adaptations to resist filovirus infection, and reveal one pathway by which a filovirus could escape from receptor control. In sum, our results support the hypothesis that bats and filoviruses have been engaged in a long-term co-evolutionary relationship, one facet of which is a molecular arms race between the viral glycoprotein and its entry receptor, NPC1.

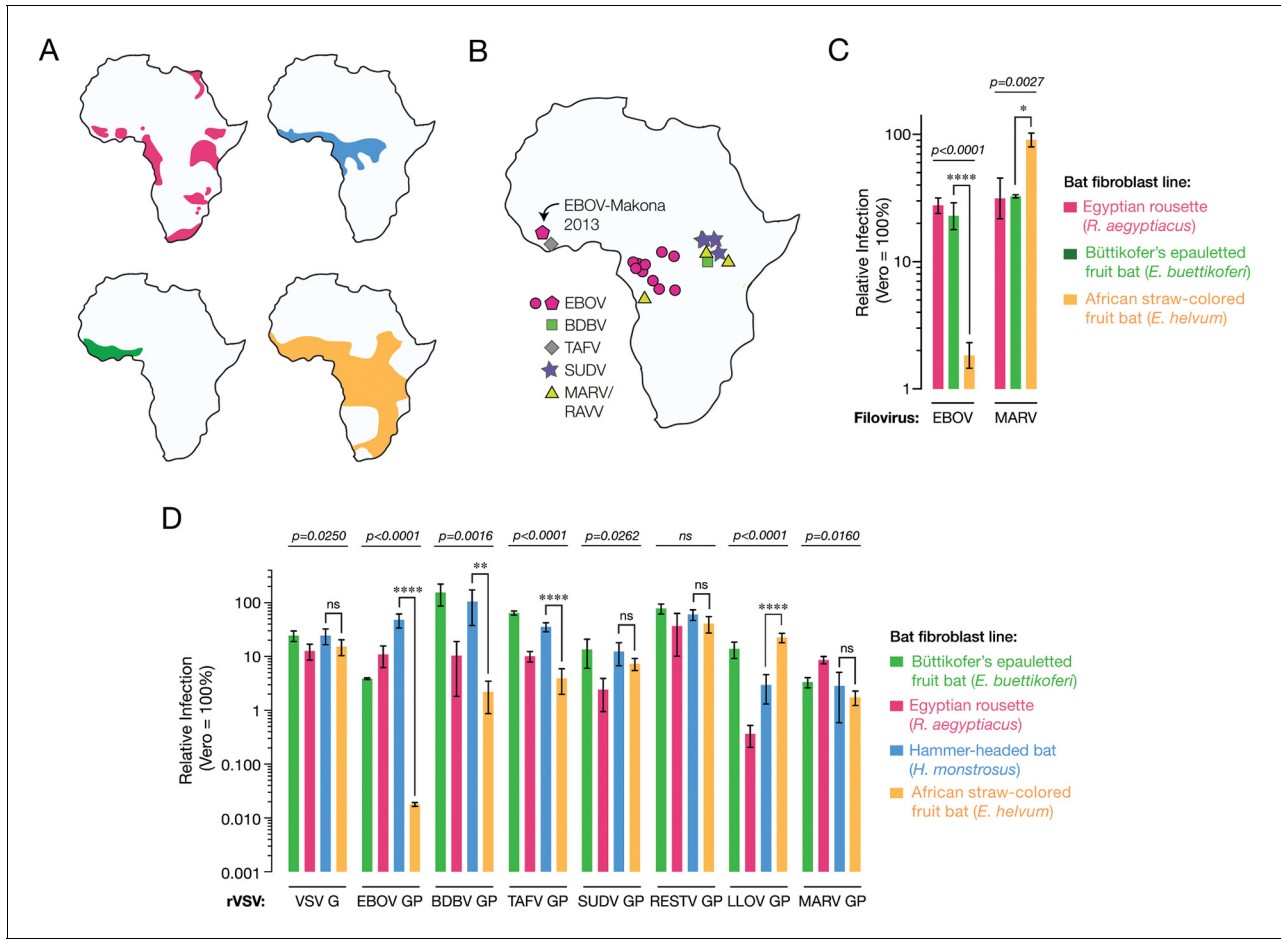

**Figure 1.** African straw-colored fruit bat cells are refractory to EBOV entry and infection. (**A**) Ranges of African pteropodids *Rousettus aegyptiacus* (pink), *Hypsignathus monstrosus* (blue), *Epomops buettikoferi* (green) and *Eidolon helvum* (yellow) (source: IUCN Redlist). (**B**) Locations of known filovirus outbreaks. (**C**) Infection of pteropodid kidney fibroblast cell lines with authentic filoviruses. Means ± standard deviations (n ≥ 3) from two biological replicates are shown. (**D**) Infections with recombinant vesicular stomatitis viruses (rVSVs) bearing filovirus glycoproteins. BDBV, Bundibugyo virus; TAFV, Taï Forest virus; SUDV, Sudan virus; RESTV, Reston virus; LLOV, Lloviu virus. Means ± SD (n = 3–4) from two biological replicates are shown. In panels C and D, the infectivity of each virus was normalized to that obtained in Vero grivet monkey cells. Means for infection of the different cell lines by each virus were compared by one-way ANOVA (p-value indicated above each group of bars). Tukey's *post hoc* test was used to compare infection means on *Hypsignathus monstrosus* vs *Eidolon helvum* cells (*p < 0.05; ****p < 0.0001; ns, no statistical significance).

The following figure supplements are available for Figure 1:

**Figure supplement 1.** Two additional African straw-colored fruit bat cell lines are selectively refractory to EBOV entry and infection.

## Results

### African straw-colored fruit bat cells are selectively refractory to EBOV infection

We first explored the possibility that there exist virus- and/or bat species-dependent differences in the cellular host range of filoviruses. Kidney fibroblast cell lines derived from three African pteropodids whose ranges overlap the locations of known African filovirus disease outbreaks (*Figure 1A,B*) were exposed to authentic EBOV and MARV (*Figure 1C*). We observed a large EBOV infection defect in African straw-colored fruit bat (*Eidolon helvum*) cells but not in cells from Büttikofer's epauletted fruit bats (*Epomops buettikoferi*) and Egyptian rousettes. By contrast, cells from bats of all three species were similarly susceptible to infection by MARV (*Figure 1C*). Thus, cells from African straw-colored fruit bats appear to be selectively refractory to EBOV infection.

### An NPC1-dependent block to cell entry accounts for the EBOV infection deficit in African straw-colored fruit bat cells

The viral spike glycoprotein, $GP_{1,2}$ (herein termed GP) mediates all steps of filovirus entry into the cytoplasm of host cells (*Lee et al., 2008*; *Miller and Chandran, 2012*). Vesicular stomatitis viruses bearing filovirus GP proteins (VSV pseudotypes) provide a highly validated surrogate system to recapitulate filovirus entry under biosafety level 2 containment (*Takada et al., 1997*; *Jangra et al., 2015*). To assess whether the EBOV infection defect in the African straw-colored fruit bat cells occurs at the viral entry step, we exposed an expanded panel of kidney fibroblast cell lines from four African pteropodids to VSV pseudotypes bearing GP spikes (VSV-GP) from seven filoviruses, including two non-African viruses, Reston virus (RESTV) and Lloviu virus (LLOV) (*Figure 1D*). As observed with authentic EBOV, VSV-EBOV GP infection was substantially reduced in the African straw-colored fruit bat cells; however, this virus could efficiently infect cells derived from the other pteropodids, including those of a proposed EBOV host, the hammer-headed fruit bat (*Hypsignathus monstrosus*) (*Leroy et al., 2005*). Strikingly, only VSVs bearing EBOV GP, and to a lesser degree, those bearing BDBV and TAFV GP, were deficient at infecting African straw-colored fruit bat fibroblasts. Similar strong but EBOV-specific reductions in infection were measured in two kidney and lung cell lines derived from additional African straw-colored fruit bats (*Figure 1—figure supplement 1*). Therefore, reduced infection of these bat cells by EBOV reflects a virus- and host species-specific restriction at the cell entry step.

We surmised that the filovirus receptor, NPC1, might explain the selective resistance of the African straw-colored fruit bat cells to EBOV entry and infection. Accordingly, we engineered these cells to stably express human NPC1 (*Hs*NPC1) (*Figure 2—figure supplements 1,2*), and then exposed them to EBOV (*Figure 2A*). Provision of *Hs*NPC1 substantially enhanced authentic EBOV infection in the African straw-colored fruit bat cells. By contrast, we found no evidence that either MARV infection in these cells, or EBOV/MARV infection in permissive Büttikofer's epauletted fruit bat cells was limited by receptor availability (*Figure 2A*). Finally, similar results were obtained with VSVs bearing filovirus glycoproteins (*Figure 2B*). Taken together, these findings indicate that EBOV infection is reduced in African straw-colored fruit bat cells because of a specific molecular incompatibility between the EBOV glycoprotein and the filovirus entry receptor.

### NPC1-dependent cell entry is reduced, but not completely eliminated, in African straw-colored fruit bat cells

Although EBOV entry and infection in African straw-colored fruit bat cells was consistently reduced to 0.1–1% relative to that in cells from the other pteropodids, we noted that infection was not completely blocked. To determine if EBOV could inefficiently infect these bat cells via an NPC1-independent mechanism, we used CRISPR/Cas9 genome engineering to derive an African straw-colored fruit bat cell line fully deficient in NPC1. We identified a single cell clone (*Eidolon helvum* NPC1–#1 [*Eh*NPC1–#1]) in which all *NPC1* alleles bore insertions or deletions (indels) at the expected site (*Figure 3A*). These indels were predicted to frameshift the NPC1 open reading frame at amino acid position 81 (*Homo sapiens Hs*NPC1 numbering), generating truncated polypeptides of 82, 83, and 109 residues that lacked the majority of the 1278-amino acid NPC1 sequence. *Eh*NPC1–#1 cells were deficient in clearance of lysosomal cholesterol, a well-established cellular function of NPC1

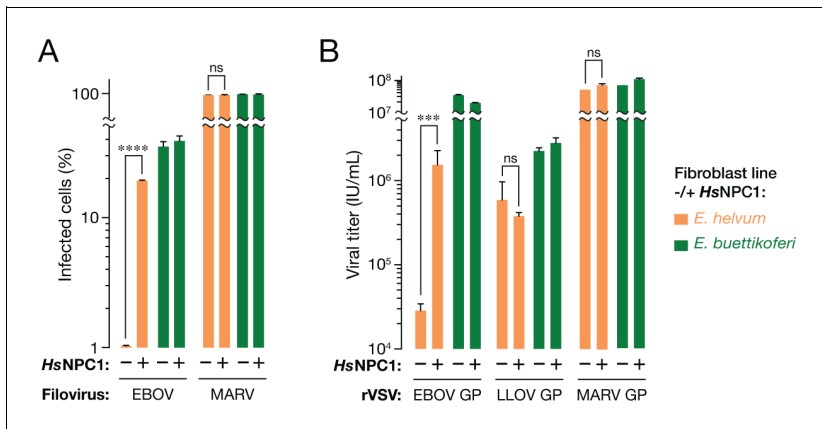

**Figure 2.** The NPC1-dependent entry and infection block in African straw-colored fruit bat cells is selective for EBOV. (**A**) Infection of pteropodid kidney fibroblast cell lines stably expressing human NPC1 (*Hs*NPC1) with authentic filoviruses. (**B**) Infection of pteropodid kidney fibroblast cell lines with recombinant VSV (rVSVs) bearing filovirus glycoproteins. IU/ml, infectious units per ml. Means ± SD (n = 3) from a representative experiment are shown in each panel. Means for infection of cell lines lacking or ectopically expressing *Hs*NPC1 were compared by unpaired two-tailed Student's t-test with Welch's correction (***p< 0.001; ****p< 0.0001; ns, no statistical significance).

The following figure supplements are available for Figure 2:

**Figure supplement 1.** Detection of endogenous NPC1 in pteropodid kidney fibroblast cell lines.

**Figure supplement 2.** Ectopic expression of human NPC1 in pteropodid kidney fibroblast cell lines.

(*Carstea et al., 1997*), but could be rescued by ectopic *Hs*NPC1 expression, confirming that NPC1 had indeed been disrupted in these cells (*Figure 3B*).

We next exposed wild-type (WT) and *Eh*NPC1–#1 fibroblasts to VSVs bearing EBOV or MARV GP. No detectable infection was obtained with either virus in NPC1-deficient cells, indicating that filovirus entry into these cells is absolutely dependent on the *E. helvum* NPC1 ortholog (*Figure 3C*). Moreover, EBOV GP-dependent infection in *Eh*NPC1–#1 cells reconstituted with *Hs*NPC1 was dramatically enhanced over that observed in WT cells, whereas MARV GP-dependent infection was rescued by *Hs*NPC1 expression to a level resembling that in WT cells (*Figure 3C*). Therefore, the low levels of EBOV infection in African straw-colored fruit bat cells likely arise from the weak, but non-zero, activity of *Eh*NPC1 as an EBOV entry receptor.

## *Eh*NPC1 is poorly recognized by EBOV GP

Filovirus GPs must directly engage the second luminal domain of NPC1, domain C, during cell entry (*Krishnan et al., 2012*; *Miller et al., 2012*). Accordingly, we postulated that the African straw-colored fruit bat NPC1 ortholog is poorly recognized by EBOV GP. To test that hypothesis, we generated and sequenced *NPC1* cDNAs from all four pteropodid cell lines. Alignment of their domain C amino acid sequences with that of *Hs*NPC1 revealed a high degree of conservation (>90%), with identical arrangements of cysteine residues and similar predicted secondary structures suggestive of a similar overall fold (*Figure 3—figure supplement 1*).

To examine GP-NPC1 binding, we engineered and expressed soluble forms of the four pteropodid NPC1 domain Cs, as described for *Hs*NPC1 (*Figure 4—figure supplement 1*) (*Miller et al., 2012*). A cleaved form of EBOV GP could capture *Hs*NPC1 domain C in an ELISA, as shown previously (*Miller et al., 2012*). EBOV GP bound with similar avidity to NPC1 domain Cs derived from Egyptian rousettes (*Ra*NPC1), hammer-headed fruit bats (*Hm*NPC1) and Büttikofer's epauletted fruit bats (*Eb*NPC1), but poorly or not at all to that of African straw-colored fruit bats (*Eh*NPC1) (*Figure 4A*). Like the infection defect in African straw-colored fruit bat cells, this receptor binding defect was selective for EBOV GP, since GPs derived from MARV and the European filovirus, LLOV (*Ng et al., 2014*), bound equivalently to all four pteropodid domain Cs (*Figure 4A*). These findings

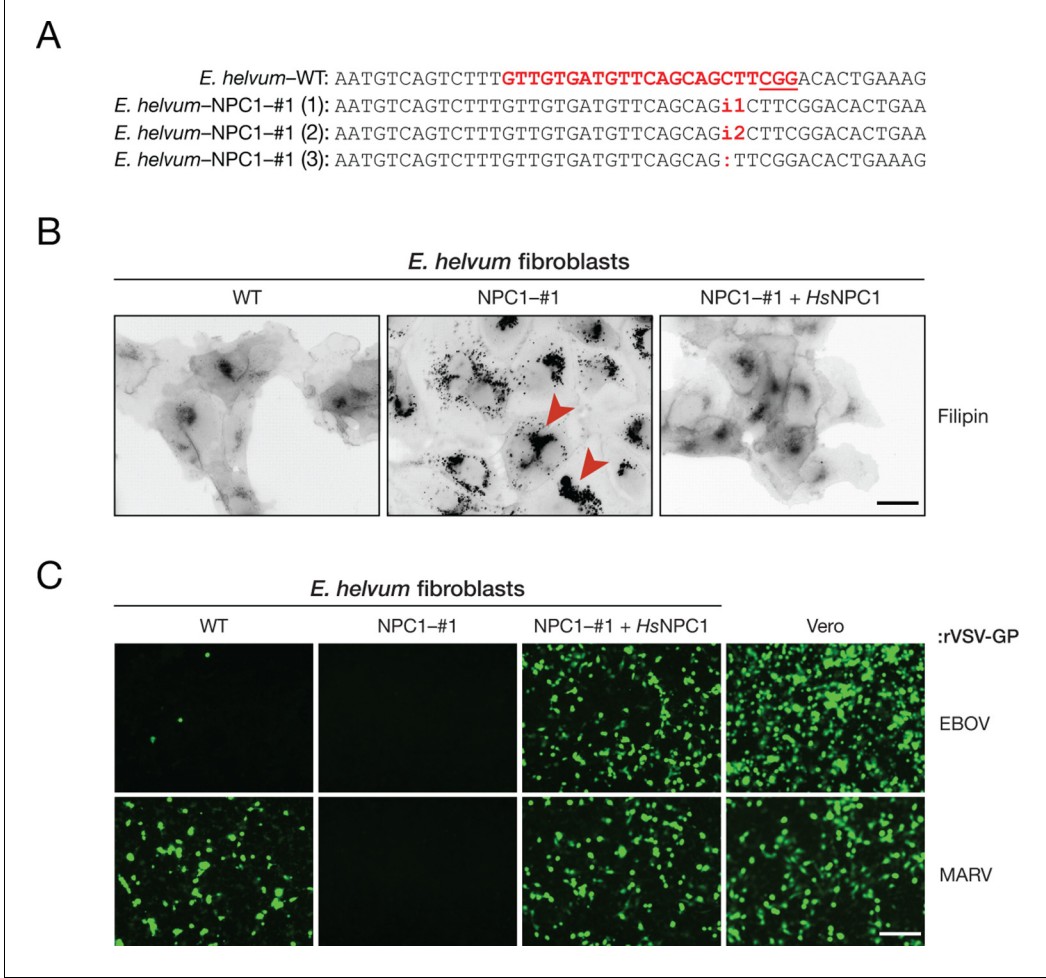

**Figure 3.** The incompatibility between EBOV GP and *Eidolon helvum* NPC1 reduces, but does not eliminate, EBOV entry into African straw-colored fruit bat cells. (A) CRISPR/Cas9 genome engineering was used to knock out the *NPC1* gene in African straw-colored fruit bat kidney fibroblasts. WT *NPC1* gene sequence aligned with the sequences of all three alleles in the knockout (NPC1–#1) cell clone. The gRNA target sequence is marked in red, and the protospacer adjacent motif (PAM) sequence of the gRNA target site is underlined. (B) The capacity of WT and NPC1–#1 cells, and NPC1–#1 cells stably expressing *Hs*NPC1, to clear lysosomal cholesterol was determined by staining with filipin III complex from *Streptomyces filipensis,* as described (*Carette et al., 2011*). Red arrowheads indicate lysosomes with accumulated cholesterol. (C) Infection of African straw-colored fruit bat cell lines and Vero African grivet monkey cells (control) by VSVs bearing EBOV or MARV GP. Infected (eGFP-positive) cells were visualized by fluorescence microscopy. Representative fields are shown. Scale bars, 20 µm.

The following figure supplements are available for Figure 3:

**Figure supplement 1.** Alignment of bat NPC1 domain C amino acid sequences.

strongly suggest that African straw-colored fruit bat cells are poorly susceptible to EBOV infection because EBOV GP poorly recognizes their ortholog of the filovirus receptor, NPC1.

## The restriction in *Eh*NPC1-EBOV GP binding can be mapped to a single amino acid change in *Eh*NPC1

To define the molecular basis of the defect in interaction between EBOV and *Eh*NPC1, we generated a panel of NPC1 domain C chimeras comprising sequences from permissive *Ra*NPC1 and non-permissive *Eh*NPC1, and tested them in the GP-binding ELISA. A single chimera, *Eh*NPC1 domain C containing four *Eh*NPC1→*Ra*NPC1 amino acid residue changes, regained the capacity to efficiently recognize EBOV GP (*Figure 4B*). Further dissection revealed that only a single amino acid change, F502D, in a central region of NPC1 domain C was needed to effect this complete restoration in GP-

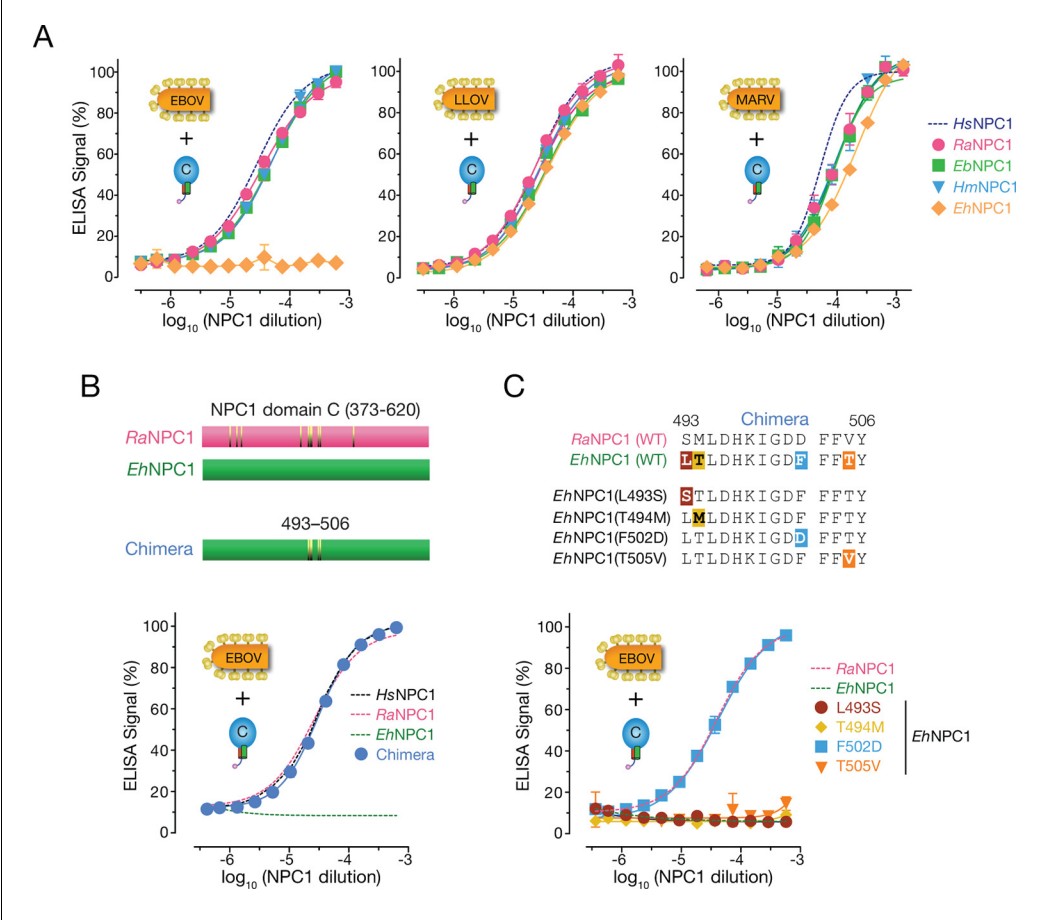

**Figure 4.** African straw-colored fruit bat NPC1 binds poorly to EBOV GP because of a single amino acid change relative to NPC1 from permissive African pteropodids. (**A**) Binding of filovirus GP proteins to soluble NPC1 domain C proteins derived from African pteropodids measured by an ELISA. *Ra*NPC1, Egyptian rousette; *Eb*NPC1, Büttikofer's epauletted fruit bat; *Hm*NPC1, Hammer-headed bat; *Eh*NPC1, African straw-colored fruit bat. (**B**) A chimera between *Ra*NPC1 and *Eh*NPC1 domain Cs fully rescues EBOV GP-*Eh*NPC1 binding. (**C**) A single amino acid change in *Eh*NPC1 domain C, F502D, renders it fully competent to recognize EBOV GP. Means ± SD (n = 3) from a representative experiment are shown.

The following figure supplements are available for Figure 4:

**Figure supplement 1.** Expression of soluble pteropodid NPC1 domain C proteins.

**Figure supplement 2.** Amino acid residue 502 is conserved in NPC1 domain C sequences from additional wild-caught African straw-colored fruit bats.

NPC1 binding (*Figure 4C*). The presence of residue F or D at position 502 for all four pteropodid NPC1 proteins tested was fully concordant with EBOV GP-NPC1 binding and NPC1 receptor function, and was also true for *Hs*NPC1 (D, permissive; F, restrictive) (*Figures 1,2*, *Figure 3—figure supplement 1*). Finally, analysis of NPC1 sequences derived from nine additional wild-caught African straw-colored fruit bats confirmed that the presence of residue F at position 502 is a conserved feature of *Eh*NPC1 (*Figure 4—figure supplement 2*). We conclude that a species-specific defect in virus-receptor interaction, caused by a single amino acid residue change in *Eh*NPC1 relative to other, permissive African pteropodid NPC1 orthologs, reduces EBOV infection in African straw-colored fruit bat cells. Moreover, because residues in the NPC1-binding site are conserved among all available EBOV GP sequences (*Supplementary file 1*), this restriction is almost certain to be encountered by all known EBOV variants and their isolates, including those detected in EBOV disease patients during the recent epidemics in Western and Middle Africa (*Park et al., 2015*; *Gire et al., 2014*; *Tong et al., 2015*; *Carroll et al., 2015*; *Kugelman et al., 2015*).

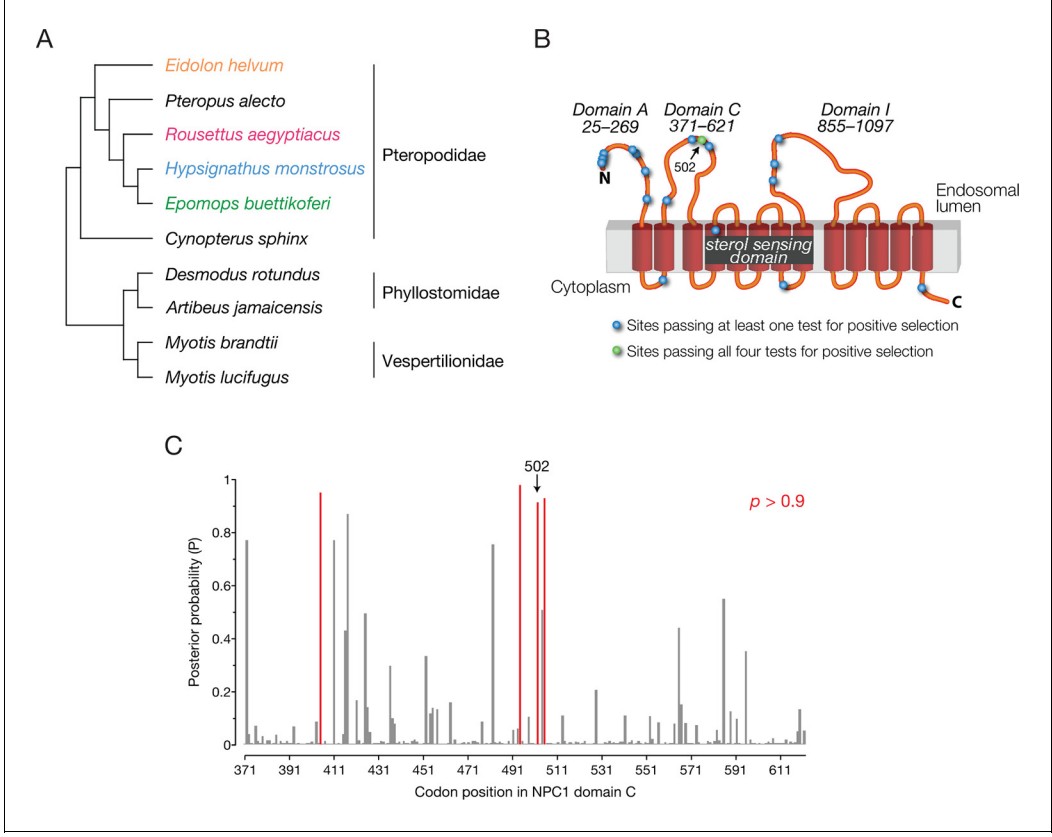

**Figure 5.** *NPC1* is under positive selection in bats. (**A**) Bat species included in the evolutionary analysis of *NPC1*. Family relationships are indicated at the right of the sequence alignment. (**B**) Positions identified with dN/dS>1 are illustrated on a cartoon schematic of NPC1. Sites in blue were identified in at least one of the four evolutionary analyses performed, and the site in green was identified in all four analyses (*Figure 5—figure supplement 1*). (**C**) The posterior probability that each codon in domain C has dN/dS>1 according to PAML. Position 502 is indicated (p = 0.921), and two clusters of sites with elevated posterior probabilities are evident.

The following figure supplements are available for Figure 5:

**Figure supplement 1.** Four tests for positive selection in bat *NPC1*.

## *NPC1* has evolved under positive selection in bats

Previous work has led to the hypothesis that bats in equatorial Africa and elsewhere harbor filoviruses (reviewed in [*Wahl-Jensen et al., 2013*]). These results, together with our findings for virus- and host species-specific differences in cellular susceptibility to filovirus infection, hinted at the possibility of a deeper co-evolutionary relationship between filoviruses and bats. One hallmark of such a relationship between a virus and its host is the evolution, under selective pressure to resist infection, of host genes encoding proviral and antiviral factors. To evaluate whether the *NPC1* gene has evolved under positive selection in bats, we combined the *NPC1* sequences obtained in this study with those of bats from six other species (two non-African pteropodids, two phyllostomids, and two vespertilionids) compiled through assembly of publicly available RNAseq data (*Figure 5*, *Figure 5— figure supplement 1*, *Supplementary files 2–4*). We then analyzed this *NPC1* multiple alignment (*Supplementary file 4*) for codon positions enriched for nonsynonymous substitutions relative to synonymous substitutions (dN/dS>1), indicative of positive, or diversifying, selection in favor of amino-acid altering mutations. Using four common tests, we found strong evidence for positive selection in bat *NPC1* (*Figure 5—figure supplement 1*).

Examination of the specific *NPC1* codons with dN/dS>1 shed light on the nature of this selective pressure—18 codons were enriched but only one codon, position 502, was identified by all four tests (*Figure 5B,C*, *Figure 5—figure supplement 1*). Strikingly, this is the same position in NPC1 at which

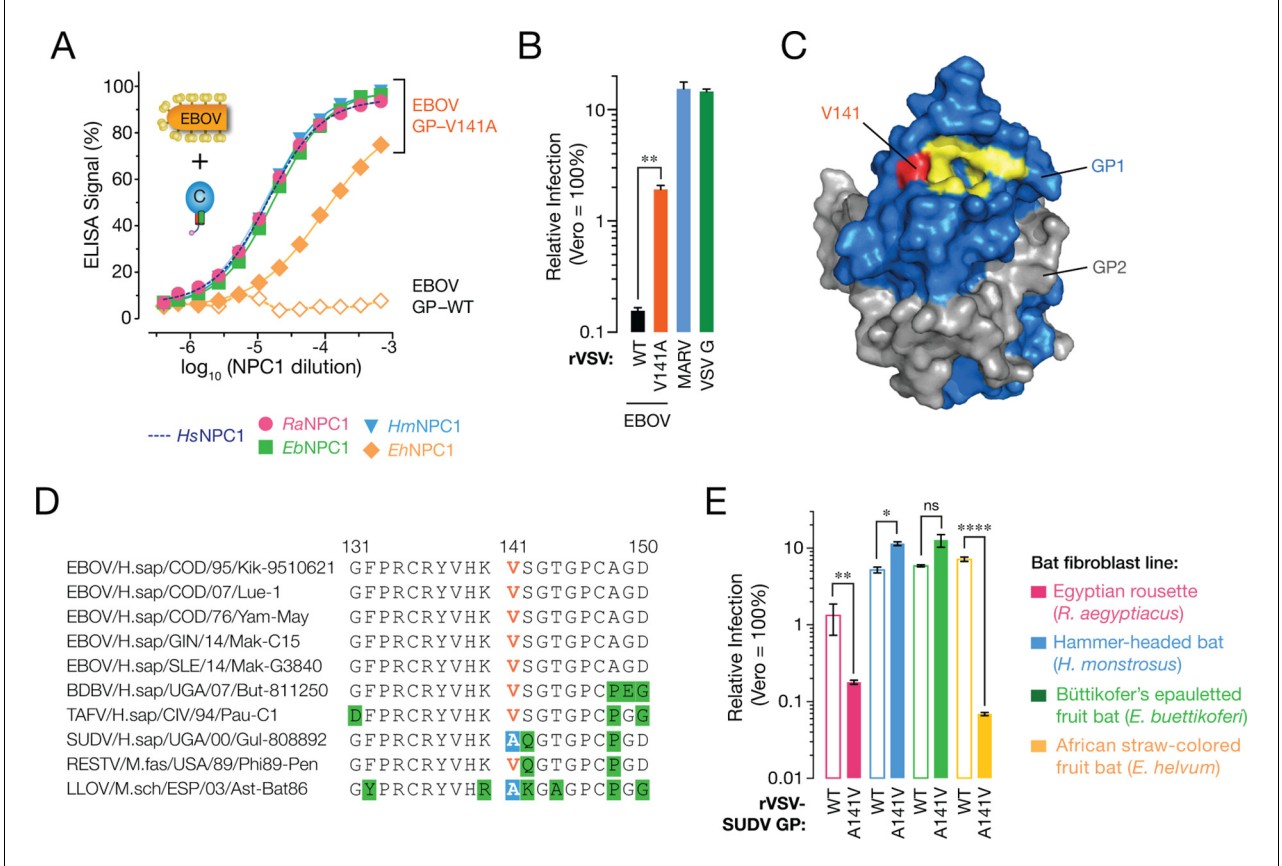

**Figure 6.** A sequence polymorphism in the NPC1-binding site of filovirus GP influences GP-*Eh*NPC1 binding and *Eh*NPC1-dependent filovirus entry. (A) Binding of EBOV GP (WT and mutant V141A) to soluble NPC1 domain C proteins derived from African pteropodids measured by an ELISA. *Ra*NPC1, Egyptian rousette; *Eb*NPC1, Büttikofer's epauletted fruit bat; *Hm*NPC1, Hammer-headed fruit bat; *Eh*NPC1, African straw-colored fruit bat. (B) Infection of African straw-colored fruit bat cells with VSV pseudotypes bearing EBOV GP (WT or V141A). Means ± SD (n = 3–4) from a representative experiment are shown in each panel. Means for VSV-EBOV GP WT vs V141A infection were compared by unpaired two-tailed Student's t-test with Welch's correction (**p < 0.01). (C) Surface-shaded representation of a single GP1-GP2 monomer (PDB ID: 3CSY (*Lee, et al., 2008*) highlighting key residues in the NPC1-binding site (yellow) and residue 141 (red). GP1, blue. GP2, grey. (D) Alignments of GP1 sequences from a panel of filoviruses. V141, orange; A141, white text on blue shading; other residues divergent from consensus sequence, black text on green shading. (E) Infection of African pteropodid cells with VSV pseudotypes bearing SUDV GP (WT or A141V). Means ± SD (n = 4) from two biological replicates are shown. Means for VSV-SUDV GP WT vs A141V infection on each cell line were compared by unpaired two-tailed Student's t-test with Welch's correction (*p < 0.05, **p < 0.01, ****p < 0.0001).

a mutation in *Eh*NPC1 reduces receptor binding to EBOV GP and viral infection, a phenotype that could reasonably produce a selective advantage (*Figure 4*). Other codons identified in only some of the tests for dN/dS>1, or at slightly lower significance levels, may still have functional significance. For example, additional codons were identified in two regions of domain C that may form a part of the recognition surface for EBOV GP (*Figure 5C*). Our finding that signatures of accelerated sequence evolution localize to structural features in NPC1 that are important for virus binding (domain C and position 502) leads us to postulate that mutations at these sites can protect bats from infection or severe disease caused by filoviruses and/or other intracellular microbes.

## A single mutation at residue 141 in EBOV GP enhances viral entry by strengthening its interaction with *Eh*NPC1

Co-evolutionary arms races between hosts and pathogens are thought to be driven by cycles of genetic adaptation and counter-adaptation (*Meyerson and Sawyer, 2011*; *Daugherty and Malik, 2012*; *Demogines et al., 2013*). In this context, we postulated that mutation of residue 502 in *Eh*NPC1 could be countered by viral mutation. To identify such putative compensatory viral changes,

we screened a panel of point mutants in the NPC1-binding site of EBOV GP by ELISA for enhanced binders to *Eh*NPC1 domain C. While no single point mutant bound to *Eh*NPC1 as well as it did to the other pteropodid NPC1s or to *Hs*NPC1, GP(V141A) partially restored *Eh*NPC1 binding (*Figure 6A*). Infection by VSV particles bearing EBOV GP(V141A) was substantially enhanced in African straw-colored fruit bat cells, commensurate with this mutant GP's increased binding affinity for *Eh*NPC1 (*Figure 6B*). Examination of the X-ray crystal structure of EBOV GP (*Lee et al., 2008*) revealed that V141 is located at the edge of the putative NPC1-binding site, where it forms part of a raised rim (*Figure 6C*). The V141A mutation likely creates a more sterically favorable (open) NPC1-binding site that can overcome the structural mismatch at the GP-NPC1 binding interface (*Figure 6C*).

### Naturally-occurring sequence variation at residue 141 in GP contributes to virus- and bat species-specific patterns of cellular susceptibility to filoviruses

Although no known EBOV isolate contains the V141A mutation, we observed that LLOV and Sudan virus (SUDV) GP naturally possess A141 (*Figure 6D*). Because both GP proteins could mediate efficient viral entry into African straw-colored fruit bat cells (*Figure 1D*) and bind to *Eh*NPC1 (*Figure 4A* data not shown for SUDV), we postulated that amino acid changes at position 141 in the GP receptor-binding site broadly influence the capacity of filovirus glycoproteins to utilize *Eh*NPC1 for viral entry. Accordingly, we exposed pteropodid kidney fibroblasts to VSV pseudotypes bearing SUDV GP(WT) or SUDV GP(A141V) (*Figure 6E*). Consistent with our hypothesis, the A141V mutation substantially reduced SUDV GP-dependent infection in African straw-colored fruit bat cells. Unexpectedly, this mutant virus also infected Egyptian rousette cells significantly less well than WT, pointing to the existence of sequence context-dependent effects that selectively affect SUDV GP(A141V) binding to *Ra*NPC1 (*Figure 6E*). These findings provide evidence that GP residue 141 can influence cellular susceptibility to infection by modulating NPC1 recognition in a manner that depends on the sequences of both proteins. We speculate that sequence variation at residue 141 and potentially other positions in the receptor-binding site of filovirus glycoproteins has been shaped by selective pressure to utilize restrictive NPC1 receptors, with potential consequences for viral host range and virulence.

## Discussion

The ongoing, unprecedented Ebola virus disease epidemic in Western Africa highlights the urgent need to uncover the biological and ecological factors that underlie the distribution, evolution, and emergence of filoviruses. While a full answer to this question will require the integration of knowledge across multiple levels of biological organization, from genes to populations to ecosystems, previous work has shown that studies of molecular interactions between viruses and their host cells can contribute important pieces to this puzzle. The essential interactions between viruses and their entry receptors provide particularly cogent examples. A switch in receptor binding from the feline to the canine ortholog of the transferrin receptor drove the emergence of a new virus, canine parvovirus, and fueled a global disease pandemic in dogs (*Allison et al., 2014*). Analyses of interactions of SARS-like coronaviruses with their receptor ACE2 have helped to trace the emergence of SARS coronavirus from bats to humans, and its use of civets as intermediate amplifying hosts (*Demogines et al., 2012*; *Ge et al., 2013*; *Ren et al., 2008*).

In this study, we show that interactions between filoviruses and their entry receptor NPC1 can influence the cellular susceptibility of bats to infection. This observation is especially striking in light of previous findings that filoviruses could efficiently infect a broad range of mammalian cells, including some derived from bats (*Kuhn, 2008*; *Kuhl et al., 2011*). Indeed, this prior work and the results of experimental infection studies in rodents and bats have led to the hypothesis that interactions between viral components and those of the host innate and adaptive immune systems constitute the primary molecular variables influencing filovirus host range in nature (*Ebihara et al., 2006*; *Volchkov et al., 2000*).

Here, we propose that *NPC1* is also a genetic determinant of filovirus susceptibility in bats. The essential nature of NPC1 for infection in cells derived from mammals of multiple species, including bats (*Figure 3*), and for infection and in vivo pathogenesis in lethal EBOV infection mouse models

argues against the existence of alternative filovirus entry receptors (*Carette et al., 2011*; *Miller et al., 2012*; *Herbert et al., 2015*). Therefore, strong reductions in the affinity of virus-NPC1 recognition are predicted to reduce or eliminate infection in whole bat hosts, as observed in *NPC1-deficient* mice (*Carette et al., 2011*; *Herbert et al., 2015*), barring viral mutation to enhance this affinity. It is conceivable that even modest defects or delays in viral multiplication through such a mechanism could help determine host range by accelerating viral immune clearance, as recently observed in *NPC1*-heterozygous mice (*Herbert et al., 2015*), or by synergizing with other host-virus barriers. The highly virus- and host species-specific nature of the virus-receptor mismatch uncovered in this study warrants the determination of more bat *NPC1* sequences for inclusion in genetic analyses (see below), and a more comprehensive phenotypic examination of virus-bat pairs. Such studies maydiscover additional interesting bat-filovirus dynamics, including incompatibilities between filoviruses and NPC1 or other proviral/antiviral host factors. Such discoveries have potential implications for our understanding of the molecular basis of filovirus infection, virulence, and host range.

We found that a single amino acid change, at residue 502, in the African straw-colored fruit bat ortholog of NPC1 (*Eh*NPC1) greatly diminished the susceptibility of cells from multiple tissues and individuals to EBOV. These migratory pteropodids are widely distributed across sub-Saharan Africa (*Figure 1A*), roost in large colonies near human settlements, and host other RNA viruses with zoonotic potential (*Baker et al., 2013*; *Peel et al., 2013*). Moreover, they are extensively hunted for bushmeat in Western Africa (*Kamins et al., 2011*), making them ideal candidates to transmit viruses directly to humans. Unfortunately, there is little information currently available on the susceptibility of African straw-colored fruit bats to EBOV or their potential role as filovirus hosts. Serologic surveys have found some evidence for exposure to one or more filovirus; however, neither infectious virus nor coding-complete or full viral genomes—the gold standards—have been successfully obtained from these bats, indicating they may only have been exposed to filoviruses, rather than being productively infected (reviewed in [*Wahl-Jensen et al., 2013*; *Olival and Hayman, 2014*]). While more extensive wildlife sampling and, if feasible, experimental infections of African straw-colored fruit bats will be required to clarify this picture, we can extrapolate to several possible scenarios. First, these bats are fully resistant to EBOV, and therefore cannot be the source of this virus in the 2013–present EBOV disease outbreak in Western Africa or the 2014 outbreak in Middle Africa. Second, because African straw-colored fruit bat cells do remain weakly susceptible to EBOV (*Figure 3C*), it is conceivable that they support EBOV replication at low levels. Indeed, this is one hallmark of a sustaining viral reservoir. Third, the filoviruses circulating in these bats, whether EBOV or otherwise, bear one or more GP mutations (e.g., V141A) that circumvent the infection barrier imposed by *Eh*NPC1. Assessing this last hypothesis and understanding the nature of the selection pressures that drive GP evolution in vivo will require the isolation of ebolavirus GP sequences from bats—there are none currently available.

Although these results suggest that African straw-colored fruit bats are selectively refractory to EBOV, our genetic findings indicate that this is not merely a special relationship between one host and one virus. Rather, we used a diverse set of bat *NPC1* sequences, only one of which is from African straw-colored fruit bats, to show that a number of codons, including residue 502, have evolved under recurrent positive selection. This is a process in which resistant *NPC1* variants are 'serially replaced' in response to compensating viral mutations that restore susceptibility. We provide evidence that the filovirus GP interaction surface in the second luminal domain of NPC1, domain C, is a hotspot for such positive selection (*Figure 5*). By contrast, the vast majority of codons in mammalian *NPC1* have evolved under purifying selection. We propose that this pattern of selection is the signature of a long-term genetic conflict between filoviruses and *NPC1* in bats, superimposed over the normal evolutionary signature of a housekeeping gene with a critical role in cellular cholesterol trafficking. Similar signatures of recurrent positive selection have been identified in other housekeeping genes that encode viral receptors, including the transferrin receptor (*Kaelber, et al., 2012*; *Demogines et al., 2013*) (TfR; receptor for New World arenaviruses [*Radoshitzky et al., 2007*], the betaretrovirus murine mammary tumor virus [*Ross et al., 2002*], and parvoviruses [*Parker et al., 2001*]), bat angiotensin-converting enzyme-2 (*Demogines et al., 2012*) (ACE2; receptor for SARS-like coronaviruses [*Li et al., 2003*]), and mammalian dipeptidyl peptidase-4 (*Cui et al., 2013*) (DPP; receptor for MERS-like coronaviruses [*Raj et al., 2013*]). In these cases as well, the preponderance of positively-selected residues localize to virus-receptor interfaces. Interestingly, the sequence polymorphism at NPC1 residue 502 did not impair cholesterol clearance from lysosomes

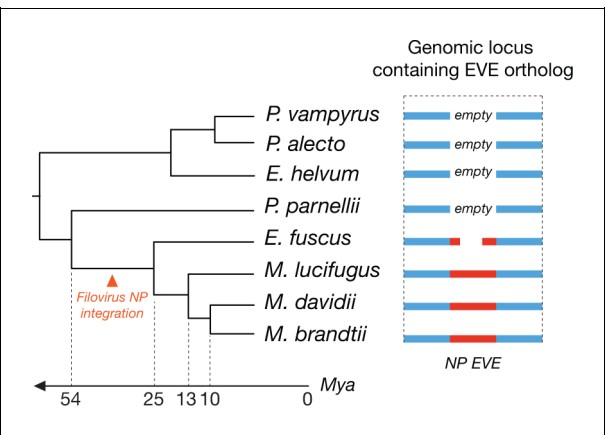

**Figure 7.** Orthologous endogenous viral elements (EVEs) derived from filovirus nucleoprotein (NP) genes indicate that filoviruses have infected bats for at least 25 million years. The time-calibrated phylogeny shown to the left is based on estimates obtained in *Miller-Butterworth et al., 2007*. The schematic to the right shows the orthologous EVEs and empty insertion sites as they occur in each bat genome. Also see *Supplementary file 5*.

(*Figure 3*), and none of the residues under positive selection were found to be mutated in Niemann-Pick type C disease patients (*Runz et al., 2008*; *Vanier and Millat, 2003*). Thus, despite being constrained by its housekeeping function, NPC1 appears to retains a sizeable sequence space accessible to adaptive mutation.

It is tempting to speculate that sequence variation at residue 141 (*Figure 6*) and potentially other positions in the receptor-binding site of filovirus glycoproteins represents the other half of the genetic arms race, shaped by selective pressure to utilize restrictive NPC1 receptors. Although more data, especially filovirus sequences from bats, are needed, our findings raise the tantalizing possibility that filoviruses, including those yet undiscovered, are each adapted to specific bat hosts, with co-evolved virus-receptor interactions constituting one potential biological barrier to interspecies viral transmission. Alternatively, it is conceivable that repeated contacts between unknown (non-bat) reservoir hosts carrying specific filoviruses, and bats of particular species, have driven positive selection in bat *NPC1* to limit infection (and selection of filoviruses with compensating sequence changes in *GP*). In this scenario, detection of anti-filovirus antibodies or filovirus genome-derived oligonucleotides may reflect a type of spillover event from the actual filovirus reservoir hosts into bats.

Our hypothesis that NPC1 in bats has been genetically sculpted by filoviruses (and vice versa) presupposes not only a long-term coevolutionary relationship, but also one in which these viruses have imposed selective pressure on bats to limit or eliminate infection. The discovery of filovirus NP- and VP35-related endogenous viral elements (EVEs) in bat genomes is consistent with such a long-term relationship (*Taylor et al., 2010*; *2011*; *Katzourakis and Gifford, 2010*). To further investigate the deeper origins of filoviruses in bats, we screened all available bat genomes for filovirus-related EVEs. We obtained evidence for synteny between a filovirus nucleoprotein (*NP*)-like EVE in the genome of the big brown bat (*Eptesicus fuscus*) and those previously identified in three, more distantly-related, myotis bats (*Figure 7* and *Supplementary file 5*) (*Taylor et al., 2011*). This new discovery strongly suggests that all four EVEs resulted from a single insertion event prior to the divergence of the *Myotis* and *Eptesicus* genera, ≈25 million years ago (*Miller-Butterworth et al., 2007*). Therefore, bats may have been exposed to filovirus-like agents for far longer than previously recognized (≈13 million years ago [*Taylor et al., 2011*]).

Available experimental exposure studies, although limited in number and scope, suggest that some filoviruses isolated from humans can replicate in bats without causing substantial host pathology (e.g., MARV and RAVV in Egyptian rousettes [*Amman et al., 2015*; *Jones et al., 2015*; *Paweska et al., 2012*]). These observations therefore prompt a key question: what is the origin and nature of the selective pressure that has driven accelerated *NPC1* evolution in bats? Our scant understanding admits a number of possibilities. First, it is conceivable that some filoviruses do

indeed replicate in a manner that is deleterious to their specific bat hosts—we may simply not have identified the viruses and hosts in question. Indeed, the filovirus LLOV, discovered in Schreibers's long-fingered bat carcasses in Spain and Portugal, may exemplify this possibility (*Negredo et al., 2011*). Alternatively, in some cases (e.g., ebolaviruses and Egyptian rousettes), the human viral isolates used in challenge studies may differ from these bat isolates in important respects due to human adaptation (human EBOV, BDBV, TAFV, RESTV, and SUDV isolates do not infect Egyptian rousettes [*Jones et al., 2015*]). Second, filoviruses may have been more virulent in bats in the past. Thus, the positive selection signatures observed in bat *NPC1*, which cannot be accurately dated, may represent fixed alleles that are the consequence of a selective process driven by ancient filoviruses with properties distinct from their modern counterparts. Indeed, the lack of virulence observed in some bats may reflect a détente that was shaped by precisely these historic genetic conflicts between filoviruses and bats. Third, we cannot rule out the (unlikely) possibility that the evolution of *NPC1* in bats was driven by an entirely different infectious agent that also utilizes (or utilized) NPC1 to multiply in its hosts. Regardless of the mechanisms that genetically shaped *NPC1*, we propose that polymorphisms in this gene nevertheless impose host barriers that impede the colonization and spread of present-day filoviruses in bats in Africa and elsewhere. Our findings set the stage for broader explorations of species-specificity in filovirus interactions with proviral and antiviral host factors, with an eye to uncovering new molecular arms races between filoviruses and bats and new genetic determinants of filovirus host range and host switching.

## Materials and methods

### Bat cells and tissues
The following immortalized pteropodid fibroblast cell lines were used: RoNi/7.1 (kidney; *Rousettus aegyptiacus*), HypNi/1.1 (kidney; *Hypsignathus monstrosus*) (*Kuhl et al., 2011*), EpoNi/22.1 (kidney; *Epomops buettikoferi*) (*Kuhl et al., 2011*), EidNi/41.2, EidNi/41.3 (kidney; *Eidolon helvum*), and EidLu/20 (lung; *Eidolon helvum*) (*Biesold et al., 2011*). The species origin of each cell line was confirmed in the publication in which it was first described (*Kuhl et al., 2011*).

Bat cell populations stably expressing human NPC1 (*Hs*NPC1) were generated as described previously (*Carette et al., 2011*). Briefly, subconfluent monolayers of cells were transduced with a retroviral vector expressing *Hs*NPC1 modified at the C–terminus with a triple flag epitope tag. Transduced cells were selected by puromycin treatment (10 µg/mL).

Licenses for capturing and export of bats, as well as ethical review and clearances of animal handling procedures were obtained from the Ghana Forestry Commission of the Ministry of Food and Agriculture. Bat organ samples were obtained as described (*Drexler et al., 2012*). Bats were caught, anesthetised with ketamine/xylazine and exsanguinated by heart puncture. Carcasses were transported on ice to a nearby laboratory facility, and organs were dissected and immediately snap-frozen for long-term storage. Animals were typed morphologically and genetically as described previously (*Kuhl et al., 2011*).

### Other cell lines
Vero African grivet kidney cells and 293T human embryonic kidney fibroblast cells were obtained from ATCC. Cell lines were maintained in Dulbecco's modified Eagle medium (DMEM) (Life Technologies, Grand Island, NY) and supplemented with 10% fetal bovine serum (Atlanta Biologicals, Flowery Branch, GA), and 1% penicillin-streptomycin (Life Technologies). All cell lines were maintained in a humidified 37°C, 5% $CO_2$ incubator.

### CRISPR/Cas9-mediated knockout of NPC1 ortholog in African straw-colored fruit bat cells
We knocked out the *NPC1* gene in the EidNi/41.3 cell line by CRISPR-Cas9-mediated genome editing as described previously (*Mali et al., 2013*). A CRISPR guide RNA (gRNA) sequence to target 5'-GTTGTGATGTTCAGCAGCTTCGG-3' in the *E. helvum* NPC1 mRNA was cloned into the gRNA cloning vector (Addgene Plasmid #41824). EidNi/41.3 cells were co-transfected with plasmid encoding human codon-optimized endonuclease Cas9 (hCas9, Addgene Plasmid #41815), gRNA cloning vector encoding the *E. helvum* NPC1-specific gRNA, a monomeric red fluorescent protein (mRFP1)

expression plasmid (to monitor transfection efficiency), and pMX-IRES-Blasti (confers blasticidin resistance to transfected cells) using Lipofectamine 2000 (Life Technologies). At 24 hr post-transfection, transfected cells were selected with 50 µg/ml of blasticidin for 24 hr and then allowed to recover in the absence of the selection agent.

Total RNA was isolated from surviving cells with the RNAeasy Mini kit (Qiagen, Valencia, CA) as per the manufacturer's directions. The *E helvum* NPC1 mRNA sequence flanking the gRNA target site was amplified with the One-Step RT-PCR kit (Qiagen) and the following primers: Forward: 5'-AT-TCTGGACTACCAAAATCTTTGCC-3', and  Reverse: 5'- ACATGGCATCCAAGCCCAAG-3'. Thermocycling conditions used for the RT-PCR were: 50°C for 30 min (reverse transcription), followed by 95°C for 15 min (initial PCR activation), then 30 cycles of 94°C for 30 sec, 60°C for 30 sec, 72°C for 1 min, then a final extension of 72°C for 10 min. Amplified PCR products were tested for indels at the target site with the SURVEYOR mutation detection kit for standard gel electrophoresis (Transgenomic, Omaha, NE), as per the manufacturer's instructions. Once indels were confirmed, amplified PCR products from single cell clones were cloned into a TOPO-TA vector (Life Technologies). Multiple clones for each single cell population were sequenced to confirm disruption of NPC1 alleles.

## VSVs and infections

Recombinant vesicular stomatitis Indiana viruses (VSVs) expressing eGFP, and EBOV, MARV, or LLOV GP in place of VSV G have been described previously (*Miller et al., 2012*; *Wong et al., 2010*; *Ng et al., 2014*). VSV pseudotypes bearing glycoproteins derived from VSV, EBOV, BDBV, TAFV, SUDV, and MARV were generated essentially as described previously (*Takada et al., 1997*). VSV particles containing $GP_{CL}$ were generated by incubating rVSV-GP-EBOV with thermolysin (200 µg/mL) (Sigma-Aldrich, St. Louis, MO) for 1 hr at 37°C. The protease was inactivated by addition of phosphoramidon (1 mM) (Sigma-Aldrich), and reaction mixtures were used immediately. Infectivities of VSV pseudotypes were measured by manual counting of eGFP-positive cells using fluorescence microscopy at 16–24 hr post-infection, as described (*Chandran et al., 2005*). Infectivities of rVSVs were measured in a similar manner, except that $NH_4Cl$ (20 mM) was added to infected cell cultures at 1–2 hr post-infection to block viral spread, and individual eGFP-positive cells were manually counted at 12–14 hr post-infection.

## Authentic filoviruses and infections

The wild-type filoviruses Ebola virus/H.sapiens-tc/COD/1995/Kikwit-9510621 (EBOV/Kik-9510621; "EBOV-Zaire 1995") and Marburg virus/H.sapiens-tc/DEU/1967/Hesse-Ci67 (MARV/Ci67) used in this study were described previously (*Jahrling et al., 1999*; *Swenson et al., 2008*). Cells were exposed to virus at an MOI of 1 pfu/cell (*Figure 1C*) or 3 pfu/cell (*Figure 2A*) for 1 hr. Viral inoculum was then removed, and fresh culture media was added. At 48 hr (*Figure 2A*) or 72 hr (*Figure 1C*) post-infection, cells were fixed with formalin and blocked with 1% bovine serum albumin (BSA). EBOV-infected cells and uninfected controls were incubated with EBOV GP-specific monoclonal antibody KZ52 (*Maruyama et al., 1999*). MARV-infected cells and uninfected controls were incubated with MARV GP-specific monoclonal antibody 9G4 (*Swenson et al., 2004*). Cells were washed with PBS prior to incubation with either goat anti-mouse IgG or goat anti-human IgG conjugated to Alexa 488. Cells were counterstained with Hoechst stain (Invitrogen, Carlsbad, CA), washed with phosphate-buffered saline (PBS), and stored at 4°C. Infected cells were quantitated by fluorescence microscopy and automated image analysis. Images were acquired at 20 fields/well with a 20× objective lens on an Operetta high content device (PerkinElmer, Waltham, NY). Operetta images were analyzed with a customized scheme built from image analysis functions available in Harmony software.

## NPC1 sequences and evolutionary analyses

From bats of four species (*Hypsignathus monstrosus*, *Eidolon helvum*, *Epomops buettikoferi,* and, *Rousettus aegyptiacus*), mRNA was collected from cell lines (or spleen samples for additional *Eidolon helvum* NPC1 domain C sequences; *Figure 4—figure supplement 2*), cDNA libraries were constructed, and the *NPC1* transcript was sequenced (see *Supplementary file 3* for primers). Using available RNAseq read data (*Supplementary file 2*), we assembled bat transcriptomes and identified *NPC1* sequences in bats of six additional species (*Myotis brandtii, Artibeus jamaicensis, Cynopterus*

*sphinx, Myotis lucifugus, Pteropus alecto,* and *Desmodus rotundus*). Transcriptome data were cleaned with Trimmomatic (*Bolger et al., 2014*) and assembled using Trinity (*Grabherr et al., 2011*) and Trans-ABySS (*Robertson et al., 2010*). The 10-species *NPC1* alignment (*Supplementary file 4*) was analyzed for positive selection using the M8 codon model in the codeml package in PAML (*Yang et al., 2000*), REL, and FEL (*Pond and Frost, 2005*), and MEME (*Murrell et al., 2012*) available at http://datamonkey.org/ (*Delport et al., 2010*). All evolutionary analyses were done using both the *NPC1* gene tree and a species tree (*Figure 5—figure supplement 1*). The species tree represents the accepted relationships amongst the bats analyzed (*Agnarsson et al., 2011*; *Almeida et al., 2011*).

## Genome screening in silico

To identify orthologous filovirus-related EVE insertions, we screened bat genomes in silico for EVEs. A representative set of filovirus protein sequences was obtained from GenBank, supplemented by the putative protein sequences of previously identified filovirus EVEs (*Taylor et al., 2014*; *Taylor et al., 2011*; *Taylor et al., 2010*; *Katzourakis and Gifford, 2010*). These sequences were used as 'probes' in tBLASTn screens of whole genome shotgun (WGS) sequence data derived from bats of ten species (*Eidolon helvum, Eptesicus fuscus, Myotis brandtii, Myotis davidii, Myotis lucifugus, Pteropus alecto, Pteropus vampyrus, Megaderma lyra, Pteronotus parnellii,* and *Rhinolophus ferrumequinum*). Statistically significant matches to filovirus probes were extracted, conceptually translated, and aligned with homologous filovirus proteins. Orthologous flanking sequences were identified by BLAST comparison of EVE-containing contigs. An alignment of the identified EVEs, along with the flanking information in the relevant bat genomes, is shown in *Supplementary file 5*.

## Generation of soluble NPC1 domain C proteins

A construct engineered to encode *Hs*NPC1 domain C (residues 372–622) flanked by sequences that form a stable, antiparallel coiled coil, and fused to a preprotrypsin signal sequence with flag and hexahistidine tags at its *N*–terminus has been described (*Deffieu and Pfeffer, 2011*; *Miller et al., 2012*). Similar constructs bearing bat NPC1 domain Cs were generated by replacing the human domain C sequence with a sequence encoding domain C from each bat NPC1 ortholog. Soluble domain C was expressed in human 293-Freestyle cells (Invitrogen) and purified from supernatants by nickel affinity chromatography, as described previously (*Miller et al., 2012*). Alternatively, cell supernatants containing soluble domain C were used directly for GP-NPC1 binding ELISAs following calibration for domain C concentration (see below).

## GP-NPC1 domain C binding ELISAs

NPC1 domain C concentrations used in the ELISAs were normalized as follows. Proteins were resolved by SDS-PAGE followed by immunoblotting with an anti-flag antibody followed by an anti-mouse Alexa-680 secondary antibody (Invitrogen). Blots were visualized using the LI-COR Odyssey Imager, and the domain C band was quantified using the LI-COR Image Studio package (LI-COR Biosciences, Lincoln, NE).

Thermolysin-cleaved VSV-EBOV GP particles were captured onto high-binding 96-well ELISA plates (Corning, Corning, NY) using KZ52, a conformation-specific anti-EBOV GP monoclonal antibody. Plates were blocked with PBS containing 3% BSA, and serial dilutions of NPC1 domain C protein were then added. Bound domain C was detected with an anti-flag antibody conjugated to horseradish peroxidase (Sigma-Aldrich) and Ultra-TMB substrate (ThermoFisher, Grand Island, NY). All binding steps were carried out at 37°C for 1 hr or at 4°C overnight. ELISAs with VSVs bearing LLOV and MARV GP were performed as above, but with the following modifications. VSV-LLOV GP particles were cleaved by incubation with a reduced concentration of thermolysin (12.5 µg/mL, 37°C, 1 hr) due to its enhanced protease sensitivity relative to ebolavirus GPs, as described (*Ng et al., 2014*). The viral envelope was then labeled with biotin using a function-spacer-lipid construct (FSL-biotin) (Sigma-Aldrich), as described previously (*Ng et al., 2014*). Biotinylated viral particles were captured onto streptavidin-coated ELISA plates (0.65 µg/mL). The remainder of the steps in the ELISA were performed as described above for VSV-EBOV GP. VSV-MARV GP particles were cleaved by incubation with trypsin (150 µg/mL, 37°C, 5 min; Sigma-Aldrich), modified as above using FSL-biotin, and captured onto streptavidin-coated magnetic beads (Spherotech, Lake Forest, IL). Beads

were then aliquotted into a 96-well round-bottomed plate for the remaining ELISA steps. PBS containing 5% nonfat dry milk was used for blocking and washing steps. Binding curves were generated by nonlinear regression analysis using Prism (4-parameter logistic equation; GraphPad Software, La Jolla, CA).

## SDS-Page and immunoblotting

To detect NPC1 in primate or bat kidney fibroblasts, whole cell lysates were prepared as previously described (*Miller et al., 2012*). Briefly, cells were washed with PBS and lysed in NTE-CHAPS buffer (10mM Tris [pH 7.5], 140mM NaCl, 1mM EDTA, 0.5% vol/vol 3-[(3-cholamidopropyl)dimethylammonio]-1- propanesulfonate) (Sigma-Aldrich) containing a protease inhibitor cocktail (Roche, Basel, Switzerland), and placed on ice for 30 min. To assist in cell lysis, cell suspensions were vortexed every 10 min, and then placed on ice for 30 min. Samples were spun at 14,000 $\times$g for 10 min, and supernatants harvested for western blot. In some experiments, proteins were deglycosylated with protein *N*–glycosidase F (New England Biolabs, Ipswich, MA) according to the manufacturer's instructions.

Proteins were resolved in 8% sodium dodecyl sulfate (SDS)-polyacrylamide gels and transferred to nitrocellulose membranes. Endogenous NPC1 was detected using an anti-Niemann Pick C1 polyclonal antibody (1:1,000 dilution; ab36983, Abcam, Cambridge, MA), followed by incubation with a donkey anti-rabbit antibody conjugated to horseradish peroxidase (1:5,000 dilution, Santa Cruz Biotechnology, Dallas, TX). Endogenous cyclin-dependent kinase 4 (CDK4; loading control) was detected with a rabbit polyclonal antibody (1:1,000 dilution; sc-260, Santa Cruz Biotechnology). Ectopic expression of *Hs*NPC1-flag was detected with an anti-flag antibody conjugated to horseradish peroxidase (Sigma-Aldrich). Bands were visualized by incubation with an enhanced chemiluminescence reagent (ThermoFisher) followed by exposure to X-ray film.

## Fluorescence microscopy and image analysis

In *Figure 3*, cells were visualized using an inverted fluorescence microscope under illumination with a 63X high-numerical aperture oil objective (*Figure 3B*) or a 10X air objective (*Figure 3C*). Images were captured with an Axiocam MRm CCD camera using AxioVision software (Zeiss USA, Thornwood, NY), and imported into Photoshop (Adobe Systems, San Jose, CA) for processing. Images were cropped, inverted (*Figure 3B*), and subjected to linear adjustment for overall brightness and contrast using the Levels tool. Developed X-ray films were digitized with a flatbed scanner and processed in Photoshop as described above.

## Statistical analysis

Statistical comparison of means among multiple independent groups was carried out by one-way analysis of variance (ANOVA) with Tukey's *post hoc* test for multiple comparisons. In some figures (see Figure Legends), an unpaired two-tailed Student's t-test with Welch's correction for unequal variances (*Ruxton, 2006*) was used for pairwise comparison of independent groups. All statistical analyses were performed in GraphPad Prism.

## Acknowledgments

We thank Tyler Krause and Cecelia Harold for technical support, and Gary Crameri, Shawn Todd, and Mary Tachedjian for help with sourcing bat cDNA for *NPC1* gene amplification. We also thank Margaret Kielian, Jack Lenz, Max Nibert, Vinayaka Prasad, DeeAnn Reeder, Nancy Simmons, and Susan Tsang for useful discussions. We thank Laura Bollinger, Integrated Research Facility at Fort Detrick, for critically editing this manuscript.

Supported by grants from the US National Institutes of Health (AI101436 to KC, GM093086 to SLS), the US Defense Threat Reduction Agency (HDTRA1-11-C-0061 to SLS, CB3948 to JMD), EU FP-7 Antigone (grant 278976) and the EBOKON Project (to CD and MAM). LFW is supported in part by an NRF-CRP grant (NRF2012NRF-CRP001-056) in Singapore. JHK performed this work as an employee of Tunnell Government Services, Inc., a subcontractor to Battelle Memorial Institute, under Battelle's prime contract with NIAID (No. HHS27220070016I). SLS is a Burroughs Wellcome Fund Investigator in the Pathogenesis of Infectious Disease. KC is additionally supported by a Harold and Muriel Block Faculty Scholarship and an Irma T. Hirschl/Monique Weill-Caulier Research Award at

the Albert Einstein College of Medicine. Opinions, conclusions, interpretations, and recommendations are those of the authors and are not necessarily endorsed by the US Department of the Army, the US Department of Defense, or the US Department of Health and Human Services.

## Additional information

### Funding

| Funder | Grant reference number | Author |
|---|---|---|
| National Institutes of Health | AI101436 | Kartik Chandran |
| Defense Threat Reduction Agency | CB3948 | John M Dye |
| European Commission | EU FP-7 Antigone | Thijn R Brummelkamp |
| Bundesministerium für Bildung und Forschung | EBOKON Project | Christian Drosten Marcel A Müller |
| National Research Foundation-Prime Minister's office, Republic of Singapore | CRP001-056 | Lin-Fa Wang |
| National Institutes of Health | GM093086 | Sara L Sawyer |
| Defense Threat Reduction Agency | HDTRA1-11-C-0061 | Sara L Sawyer |

The funders had no role in study design, data collection and interpretation, or the decision to submit the work for publication.

### Author contributions

MN, EN, MEK, JMD, SLS, KC, Conception and design, Acquisition of data, Analysis and interpretation of data, Drafting or revising the article; ASH, RJG, Acquisition of data, Analysis and interpretation of data, Drafting or revising the article; TB, AIK, RMJ, Acquisition of data, Analysis and interpretation of data; RKJ, Acquisition of data, Analysis and interpretation of data, Drafting or revising the article, Contributed unpublished essential data or reagents; JAH, MY, Acquisition of data, Analysis and interpretation of data, Contributed unpublished essential data or reagents; RB, Acquisition of data, Contributed unpublished essential data or reagents; AD, Analysis and interpretation of data, Drafting or revising the article; TRB, Conception and design, Analysis and interpretation of data, Drafting or revising the article; CD, MAM, Drafting or revising the article, Contributed unpublished essential data or reagents; LFW, JHK, Analysis and interpretation of data, Drafting or revising the article, Contributed unpublished essential data or reagents

### Author ORCIDs

Jens H Kuhn, http://orcid.org/0000-0002-7800-6045
Kartik Chandran, http://orcid.org/0000-0003-0232-7077

### Ethics

Animal experimentation: Capturing and sampling of *E. helvum* bats was done with permission from the Wildlife Division, Forestry Commission, Accra, Ghana. Geographic co-ordinates of the sampling site in Kumasi/Ghana were N06u42902.00W001u37929.90. Under the auspices of Ghana authorities, bats were caught with mist nets, anaesthetized with a ketamine/xylazine mixture and euthanized by cervical dislocation (permit no. CHRPE49/09; A04957). Veterinary skilled staff performed all procedures on the bats. Additional export permission was obtained from the Veterinary Services of the Ghana Ministry of Food and Agriculture (permit no. CHRPE49/09; A04957).

## Additional files

### Supplementary files

• Supplementary file 1. The sequence of the receptor-binding site in GP is highly conserved among EBOV isolates. Alignment of GP amino acid sequences corresponding to the NPC1-binding site

(residues 53-200) derived from diverse EBOV isolates (listed GenBank accession numbers) is shown. Amino acid changes are highlighted in pink.

• Supplementary file 2. Sources of RNAseq data used to assemble *NPC1* sequences.

• Supplementary file 3. *NPC1* sequencing strategy and primers.

• Supplementary file 4. *NPC1* nucleotide sequences determined experimentally or assembled from publicly available RNAseq data.

• Supplementary file 5. Alignment of sequences flanking orthologous filovirus-related EVEs and empty insertion sites. Multiple sequence alignment showing synteny between sequences flanking filovirus *NP*-derived EVEs in the indicated bat genomes. (A) The region flanking the 5′ end of the EVE ortholog. (B) Amino acid sequence alignment of a conserved region within exogenous filovirus *NP* and the putative EVE ORFs found of these sequences (note: the *E. helvum* sequence in deleted in this region and is not shown). (C) Region flanking the 3′ end of the EVE ortholog. Nucleotide insertions that were present in only one sequence are not shown. EVE sequence is highlighted in grey. Accession numbers for EVE sequences are as follows; *M. brandtii* (ANKR01230743.1); *M. lucifugus* (AAPE02014310.1); *M. davidii* (ALWT01026193.1); *E. fuscus* (ALEH01076399.1); *E. helvum* (AWHC01132512.1); *P. vampyrus* (ABRP02039678.1); *P. alecto* (ALWS01163349.1); *P. parnellii* (AWGZ01223755.1).

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
