## [Decision Letter]

Thank you for submitting your work entitled "NPC1 contributes to species-specific patterns of filovirus susceptibility in bats" for consideration by *eLife*. Your article has been reviewed by three peer reviewers, one of whom is a member of our board of Reviewing Editors. The evaluation has been overseen by Richard Losick as the Senior Editor. The following reviewers have agreed to reveal their identity: Robert Davey and Sergei Kosakovsky Pond.

The reviewers have discussed the reviews with one another and the Reviewing editor has drafted this decision to help you prepare a revised submission.

Summary:

All reviewers agreed that you have presented a comprehensive study of the interaction between Ebola virus GP and the NPC1 in bats. Your data suggest an ancient coevolution between NPC1 in bats and the filoviruses, and you identify mutations in NPC1 that mediate host specificity. The paper further shows that mutations in EBOV GP can restore infectivity in otherwise resistant cells.

Essential revisions:

The primary issue that we ask you to address before we can recommend publication in *eLife* concerns the lack of description of statistical procedures. Please describe the degree of replication, the statistical tests, etc. Image analysis also requires a more detailed description.

Minor points:

While reviewing the manuscript, we have noticed a number of other points given below that you should consider in your efforts to improve the manuscript. Furthermore, we would like to encourage you to provide additional source files such as the alignments and tabular files with the values that go into the bar graphs. Those can be linked to figures as source files in an *eLife* paper and facilitate reproduction of your results and further analysis of the data.

Reviewer #1 (Minor Comments):

I would like to discourage the use of bar graphs. Look here for a good explanation why: http://journals.plos.org/plosbiology/article?id=10.1371/journal.pbio.1002128

17 comments on PubPeer

17 comments on PubPeer

17 comments on PubPeer

17 comments on PubPeer

17 comments on PubPeer

17 comments on PubPeer

Figure 1: don't break the axis, show on log scale instead.

Please elaborate a little on why VSV with EBOV GP is a good model for EBOV entry – for non-virologists like myself.

Reviewer #2 (Minor Comments):

Selection analyses: It would be interesting to see whether or not signatures of positive selection are localized to specific lineages, as MEME results (much stronger signal than the other three methods) would indicate, and whether or not these are the expected lineages. The authors could use branch site models (e.g. aBSREL available on test.datamonkey.org) to perform these analyses.

Selection analyses: Can a corresponding selection analysis be run on filovirus GP orthologs to identify possible "escape" mutations there (e.g. 141)?

Discussion: Move some of the EVE text to Results (and mention Figure 7 there).

Discussion: Given the low apparent genetic barrier to viral/host escape (e.g. a single point mutation, which doesn't abrogate other functions), to acquiring new host specificity, which is in contrast with more polyallelic adaptation in some other virus/host systems (e.g. lentiviruses, as shown repeatedly by the Malik lab and the Sawyer lab), what can be said about the disease and population dynamics here? Can one infer that (at least recently) the selective pressure exerted by filoviruses on NPC1 in bats is minor?

Reviewer #3 (Minor Comments):

Please take a look at Figure 6. The labeling on the origin of the constructs is a little difficult to follow. This is a minor point and does not detract from the work but would help readers to follow the data.

Reviewer #3 (Additional data files and statistical comments):

The supplemental data are excellent additions to the work. The data appears to have been statistically analyzed but the test being used (should be ANOVA for multiple groups) is not indicated. Values for the significance levels in the graphs are not given.

---

## [Author Response]

*Essential revisions:*

*The primary issue that we ask you to address before we can recommend publication in* eLife *concerns the lack of description of statistical procedures. Please describe the degree of replication, the statistical tests, etc. Image analysis also requires a more detailed description.*

The revised manuscript now includes a detailed description of methods for statistical and image analysis (new sections added to the Materials and methods). Furthermore, each figure legend now contains information on the number of biological and technical replicates, the statistical tests performed, and values for the significance levels in the graphs.

*Reviewer #1 (Minor Comments):*

*Figure 1 don't break the axis, show on log scale instead.*

Figure 1 has been replotted on a log scale.

*Please elaborate a little on why VSV with EBOV GP is a good model for EBOV entry – for non-virologists like myself.*

We have added a sentence to the Results section (subheading “An NPC1-dependent block to cell entry accounts for the EBOV infection deficit in African straw-colored fruit bat cells”) describing why VSV-GPs provide good models for filovirus entry.

*Reviewer #2 (Minor Comments): Selection analyses: It would be interesting to see whether or not signatures of positive selection are localized to specific lineages, as MEME results (much stronger signal than the other three methods) would indicate, and whether or not these are the expected lineages. The authors could use branch site models (e.g. aBSREL available on test.datamonkey.org) to perform these analyses.*

As suggested, we have now run the aBSREL test. aBSREL found no evidence for branch-wise omega variation. For this reason, there were no branches identified with episodic diversifying selection. These results may differ upon expansion of our dataset to include NPC1 from more bat species, which we plan to do as part of future studies.

*Selection analyses: Can a corresponding selection analysis be run on filovirus GP orthologs to identify possible "escape" mutations there (e.g. 141)?*

Yes, we have performed a selection analysis on filovirus GP orthologs. We chose not to include it in the paper for several reasons. First and foremost, all available GP sequences for Ebola virus are from human isolates. This reflects the fundamental problem where Ebola virus has never been isolated from its reservoir species. Therefore, our ability to study the evolutionary dynamics in this reservoir is limited.

We did run an analysis using available human-derived sequences, and we found a number of sites that were under positive selection, most of them residing in the “mucin” domain. This domain resides on the outside of the GP, and is predicted to have a multitude of functions, one of which is protecting the receptor binding site from immune detection. Therefore, we would expect to see rapidly evolving sites in this domain as a result of immune evasion. In support of this, the Sabeti group (Cell, June 2015, Figure 4) recently found a similar signal between and within outbreaks in the mucin domain, and was able to map epitopes of known antibodies to some of their positively selected sites. For this reason, we think these signatures have more to do with antibody escape as the virus emerges into the human population than in receptor binding adaptation that may be happening over deeper timescales.

When we removed the mucin domain from the alignment, a few new sites did appear with dN/dS >1, one of which (site 143) was adjacent to the predicted receptor binding region and to site 141. However, this site only showed up in one of our analyses (REL). Again, this human-derived dataset is not ideal for the question being posed, and the mucin domain finding suggests that we are primarily detecting the signals of immune evasion once the virus has entered the human population.

*Discussion: Given the low apparent genetic barrier to viral/host escape (e.g. a single point mutation, which doesn't abrogate other functions), to acquiring new host specificity, which is in contrast with more polyallelic adaptation in some other virus/host systems (e.g. lentiviruses, as shown repeatedly by the Malik lab and the Sawyer lab), what can be said about the disease and population dynamics here? Can one infer that (at least recently) the selective pressure exerted by filoviruses on NPC1 in bats is minor?*

Yes, this is quite interesting: The *E. helvum* cells can apparently acquire resistance with 1 amino acid change in 1 protein, and in turn the virus can counter adapt with a single amino acid change. This is in contrast to retroviruses where there tend to be multiple genetic blocks even in experimental infections between more closely related species (primate to primate) than what is being analyzed here (human to bat).

If Ebola viruses have the same type of devastating effect on naïve bat populations as they do on human populations, these viruses do not have an ideal evolutionary strategy for long-term survival. Perhaps a single substitution in NPC1 that allows low, yet persistent infection (i.e. a “slow burn”) is an evolutionarily stable strategy for both the virus and for the host. Our data suggest that the virus could easily adapt to overcome this receptor barrier. However, when such viral variants arise they may quickly go to extinction, compared to the slow-burn variants which go on to persist in this reservoir. This hypothesis is not as far-fetched as it may seem – the case of Hendra virus is similar to that of Ebola virus (Plowright et al., Proc. R. Soc. B 282: 20142124). Isolation of intact virus from nature is difficult, yet bats harbor antibodies against Hendra virus. Outbreaks only occur occasionally and seasonally (called “shedding pulses”), and have now been correlated to a number of events including pregnancy and nutritional stress. It is possible that both Hendra virus and Ebola virus exist at low (and hard to detect) levels in their reservoir hosts, only replicating to higher levels when certain ecological and seasonal changes occur.

Reviewer #3 (Minor Comments):

*Please take a look at Figure 6. The labeling on the origin of the constructs is a little difficult to follow. This is a minor point and does not detract from the work but would help the reader to follow the data.*

We have amended the legend to Figure 6 to clarify the origin of the constructs.

Reviewer #3 (Additional data files and statistical comments):

*The supplemental data are excellent additions to the work. The data appears to have been statistically analyzed but the test being used (should be ANOVA for multiple groups) is not indicated. Values for the significance levels in the graphs are not given.*

We have addressed these issues as part of our essential revisions